# Eggplant Germination is Promoted by Hydrogen Peroxide and Temperature in an Independent but Overlapping Manner

**DOI:** 10.3390/molecules24234270

**Published:** 2019-11-23

**Authors:** Hana Dufková, Miroslav Berka, Markéta Luklová, Aaron M. Rashotte, Břetislav Brzobohatý, Martin Černý

**Affiliations:** 1Department of Molecular Biology and Radiobiology, Faculty of AgriSciences, Mendel University in Brno, Zemedelska 1, CZ-61300 Brno, Czech Republic; dufkova.ha1@gmail.com (H.D.); miroslavberka94@gmail.com (M.B.); luklovam@gmail.com (M.L.); rashotte@auburn.edu (A.M.R.); brzoboha@ibp.cz (B.B.); 2101 Rouse Life Sciences, Department of Biological Sciences, Auburn University, Auburn, AL 36849, USA; 3CEITEC—Central European Institute of Technology, Faculty of AgriSciences Mendel University in Brno, 613 00 Brno, Czech Republic; 4Institute of Biophysics AS CR, 613 00 Brno, Czech Republic

**Keywords:** hydrogen peroxide, germination, proteomics, temperature, eggplant, seed

## Abstract

Hydrogen peroxide promotes seed germination, but the molecular mechanisms underlying this process are unclear. This study presents the results of eggplant (*Solanum melongena*) germination analyses conducted at two different temperatures and follows the effect of hydrogen peroxide treatment on seed germination and the seed proteome. Hydrogen peroxide was found to promote eggplant germination in a way not dissimilar to that of increased temperature stimuli. LC–MS profiling detected 729 protein families, 77 of which responded to a temperature increase or hydrogen peroxide treatment. These differentially abundant proteins were found to be involved in a number of processes, including protein and amino acid metabolism, carbohydrate metabolism, and the glyoxylate cycle. There was a very low overlap between hydrogen peroxide and temperature-responsive proteins, highlighting the differences behind the seemingly similar outcomes. Furthermore, the observed changes from the seed proteome indicate that hydrogen peroxide treatment diminished the seed endogenous hydrogen peroxide pool and that a part of manifested positive hydrogen peroxide effect might be related to altered sensitivity to abscisic acid.

## 1. Introduction

Hydrogen peroxide at high concentrations is known to induce oxidative damage to biomolecules; in contrast, concentrations in the low nanomolar range resemble the effects of phytohormones in many aspects [1,2,3]. Imbibed and germinating seeds produce high levels of hydrogen peroxide as a byproduct of intensive metabolism that can have a negative toxic effect but can also serve to promote gibberellin biosynthesis, decrease abscisic acid levels (e.g., [4,5,6,7]), and facilitate endosperm weakening [8]. It has been well established that hydrogen peroxide participates in the regulation of germination, but the molecular mechanisms are far from being resolved. Eggplant (*Solanum melongena*) is a perennial plant grown worldwide for its fruit, and its production has been steadily rising, reaching 52 million tons in 2017 (www.fao.org). The plant species is believed to have originated in tropical regions, and its seeds thus require a warm temperature for germination and are characterized by a relatively slow germination rate. Temperature ranges for eggplant seed germination are from 20 to 32 °C, with the highest germination rates recorded at 27–30 °C [9,10]. Eggplant seeds are characterized by a low seed germination, thus presenting an ideal model for an analysis of germination promoting substances. Furthermore, the eggplant genome is available [11], facilitating unrestricted access for “omics” approaches, including proteomics analyses. In contrast to other model species that have been previously employed in molecular analyses of germination, the size of an eggplant seed and its low germination rate allows for an easy examination of germination and facilitates a highly reproducible selection of uniform populations for molecular analyses. Our previous bioinformatics analysis revealed a positive correlation between the expression patterns of hydrogen peroxide metabolism genes and temperature-responsive genes [3]. We hypothesized that hydrogen peroxide could promote eggplant seed germination and could possibly overcome low germination at suboptimal temperatures. Here, we analyzed the influence of temperature and hydrogen peroxide on eggplant seed germination rates, and we present results of an LC–MS proteomics analysis.

## 2. Results

### 2.1. Hydrogen Peroxide Stimulates Eggplant Seed Germination

In order to test the effect of hydrogen peroxide on eggplant germination rates, seeds were imbibed in water supplemented with a range of hydrogen peroxide concentrations from 0–3000 mM at 25 °C, a suboptimal germination temperature. As expected, at 72 h after imbibition (HAI), only low levels of germination of about 10% were found without hydrogen peroxide treatment (Figure 1). At lower concentrations of 10–100 mM, hydrogen peroxide was found to promote seed germination, but higher dosages were inhibitive, including 1000 and 3000 mM, where germination was fully inhibited (Figure 1). Concentrations above 1000 mM also resulted in a visible seed discoloring (Figure 1).

### 2.2. Hydrogen Peroxide-Induced Germination Rate does not Manifest Any Visible Seedlings Traits

As the hydrogen peroxide treatment at 50 mM concentration had the highest promotion of the germination rate without visible changes in seeds compared to mock-germinated seeds, this treatment was used as a set level for further examination. In order to examine if there were any morphological changes to seedlings from lower levels of the hydrogen peroxide treatment, a homogeneous set of germinated seeds collected at 72 h after imbibition (HAI), either with 0 or 50 mM hydrogen peroxide, were transferred into soil and placed in a growth chamber at 25 °C with a 16 h photoperiod (PPFD photosynthetic photon flux density of 100 µmol m^−2^ s^−1^ over the waveband of 400–700 nm). After eight days, the seedlings were visually examined, and their cotyledon area was measured. There were no obvious visual differences in seedlings and no significant differences in the cotyledon area (Figure 2). This indicates that a 50 mM hydrogen peroxide treatment can stimulate seed germination but does not appear to impact seedling growth.

### 2.3. Hydrogen Peroxide Seemingly Mimics the Temperature Stimulus

Since hydrogen peroxide could be found to increase germination, this effect was compared to changes in germination rates by temperature. First, a set of germination assays that observed eggplant germination at 20, 25, and 29 °C were conducted. No seed germination was observed at 20 °C, and germination rates were low at 25 °C (10%) and higher at 29 °C (21%) after 72 h of imbibition. Then, treatments of 50 mM hydrogen peroxide were added to seeds germinated at 25 and 29 °C to determine if the hydrogen peroxide could stimulate germination along with temperature. This response was similar in eight independent biological replicates (Figure 3). It was observed that the positive effect of 50 mM hydrogen peroxide was similar to germination at an increased in temperature by five degrees Celsius. A statistical analysis confirmed that there was no significant difference between the number of germinated seeds at 29 and 25 °C with 50 mM hydrogen peroxide. Interestingly, the addition of 50 mM hydrogen peroxide at 29 °C further promoted germination rates, reaching, on average, 31% germinated seeds at 72 HAI, although the observed variability between the biological replicates was higher.

### 2.4. Proteomics Analysis

The observed similarity between temperature stimuli and the hydrogen peroxide treatment prompted us to perform molecular analyses. For each experimental replicate, at least ten germinated seeds 72 HAI were collected, flash-frozen in liquid nitrogen, and homogenized using a Retsch mill. Proteins were extracted by acetone/trichloroacetic acid/phenol extraction, as described previously [12], and the resulting tryptic digests were analyzed via LC–MS. Altogether, 729 protein families (representing 2678 peptide groups and 1243 proteins) were identified with the sufficient and reproducible quantitative results in at least two biological replicates for more than 300 of these. Next, a principal component analysis (PCA) analysis was performed (Figure 4) to evaluate distinct similarities in the analyzed proteomes. On average, PCA separated temperatures and hydrogen peroxide treatments in PC1 and PC2, respectively. The results did not imply any significant differences between peroxide and mock-treated seeds at 25 °C, which is in line with the results of our germination assays. Quantitative differences were detected via a spectral counting-based method by comparing (i) hydrogen peroxide and mock-treated samples, and (ii) samples cultivated at 29 and 25 °C. Only proteins with statistically significant (*p* < 0.05) relative abundance changes were selected for further analysis. The resulting list of 77 differentially abundant proteins is presented in Figure 5. In total, we found 63 and 32 hydrogen peroxide and temperature-responsive proteins, respectively (Figure 5c). The identified proteins represent a snapshot of diverse processes in seed germination, including primary metabolism (carbohydrate metabolism, amino acid metabolism, and the citrate cycle), the glyoxylate cycle or glutathione metabolism. For the list of peptides and information about all identified proteins, see the Appendix A.

### 2.5. Response to Hydrogen Peroxide and Temperature is Predominantly Different

Altogether, nearly half, or 14 out of 32, of the identified temperature-responsive proteins did not show any significant response to hydrogen peroxide. Four and ten proteins were accumulated in response to temperature but depleted in response to hydrogen peroxide at 25 and 29 °C, respectively. The comparison between responsive proteins at 25 and 29 °C showed that about half of the hydrogen peroxide-elicited responses were temperature independent, and 22 and nine were found only at 29 and 25 °C, respectively. Only five proteins showed a similar response to temperature increase and to the hydrogen peroxide treatment, including four ribosomal proteins and a protein with the conserved hydroxyacylglutathione hydrolase domain that shared a similarity with *Arabidopsis* persulfide dioxygenase (SMEL_007g294860.1.01). For details, see Figure 5 and Appendix A.

### 2.6. Expected Localization of Identified Proteins

A search for *Arabidopsis* orthologues of identified eggplant proteins was conducted, and the SUBcellular localisation database for Arabidopsis proteins SUBA4 [13] was used to estimate their subcellular distribution. Altogether, we were able to obtain putative localizations for all responsive proteins and for more than 630 of all identified eggplant seed proteins. The estimation revealed that the seed proteins were predominantly localized into cytosol (35%), plastids (16%), nuclei (11%), and extracellular spaces (11%). Putative localizations for the set of temperature and hydrogen peroxide-responsive proteins were significantly enriched in cytosolic proteins (up to a 1.5-fold increase) and depleted in plastidial proteins (up to a 2.9-fold increase).

### 2.7. Enzymes of Hydrogen Peroxide Metabolism

We hypothesized that the hydrogen peroxide treatment would have an impact on the eggplant seed’s endogenous hydrogen peroxide metabolism. Thus, a search for the major plant hydrogen peroxide metabolism enzymes in our dataset was performed. First, a BLAST search was employed, identifying 94 orthologues encoded by the eggplant genome (see Appendix A for details). However, only eight of these proteins were found in the proteomics data, including two catalases, two L-ascorbate peroxidases, sulfite oxidase, indole-3-acetaldehyde oxidase, acyl-coenzyme A oxidase, and superoxide dismutase. Surprisingly, the only significant change that was determined was the abundance of catalases that were depleted in response to hydrogen peroxide (Figure 5). The sequence analysis indicated that these catalases (SMEL_005g238230.1.01 and SMEL_004g223080.1.01) are orthologues to *Arabidopsis* catalase 2, the fundamental enzyme in basal hydrogen peroxide processing in plants [14].

### 2.8. Putative Function of Identified Differentially Abundant Proteins

To shed light into hydrogen peroxide and temperature-mediated processes during eggplant seed germination, putative interactions and key metabolic pathways represented by the set of identified differentially abundant proteins were visualized (Figure 6). The analysis revealed several functional clusters. Ribosome constituents formed the largest cluster, with 17 proteins represented in our dataset. Ribosomal proteins were mostly accumulated in response to 29 °C (12 ribosomal proteins). The hydrogen peroxide treatment resulted in a decrease in an abundance for four of these and depleted five and three additional ribosomal proteins at 25 and 29 °C, respectively. The second-largest cluster was formed by an overlapping set of enzymes involved in the amino acid biosynthesis and enzymes of glyoxylate metabolism and the citric acid cycle. The abundance of these enzymes was predominantly decreased in response to hydrogen peroxide and increased in response to 29 °C, and a similar response was found for three enzymes involved in purine metabolism (nucleoside diphosphate kinases SMEL_000g108690.1.01 and SMEL_001g132570.1.01; adenylate kinase SMEL_003g188330.1.01). The set of 17.8 kDa heat shock proteins (SMEL_008g314590.1.01, SMEL_006g268200.1.01, and SMEL_006g268190.1.01) and heat shock 70 kDa proteins (SMEL_009g330840.1.01 and SMEL_007g271490.1.01) was affected only by the hydrogen peroxide treatment. The last large cluster was formed by five enzymes that are associated with glutathione metabolism (SMEL_010g356620.1.01, SMEL_005g229410.1.01, SMEL_000g060020.1.01, SMEL_005g234070.1.01, and SMEL_006g242570.1.01). With the single exception of putative peroxisomal glutathione S-transferase, all these enzymes were depleted in response to the hydrogen peroxide treatment.

## 3. Discussion

### 3.1. Hydrogen Peroxide—A Potent Enhancer of Eggplant Germination

The eggplant is an important vegetable crop, and multiple studies have been conducted to improve its low germination rate and optimize its production. Eggplant seed dormancy has been alleviated by gibberellins and potassium nitrate [16,17,18], the application of garlic extracts [19], and with organic fertilizers [20]. Here, we demonstrated that hydrogen peroxide significantly improves the eggplant seed germination rate, presenting a cheap and ecological alternative to the previously described approaches.

### 3.2. Omics Analyses Indicate a Low Effect of Temperature on Hydrogen Peroxide Metabolism and a Possible Effect of Hydrogen Peroxide on Temperature Perception

Hydrogen peroxide plays a role in cold acclimation and thermotolerance [21,22], and our previous bioinformatics analysis revealed that hydrogen peroxide metabolism genes under stress conditions have expression patterns similar to those of genes related to temperature stress [3]. However, the documented direct effect of temperature on hydrogen peroxide metabolism genes is limited. The analysis of available *Arabidopsis* transcriptomics data by Genevestigator revealed only a few correlations between temperature stimulus and the gene expression of hydrogen peroxide metabolism genes in plants, namely ascorbate peroxidase APX2 (positive correlation) and peroxidases PER16, 17, 21, 62 and 71 (negative correlations). Thus, besides a positive effect on enzyme catalysis, the available literature does not seem to support the possibility that molecular processes promoting eggplant germination rates at 29 °C would directly recruit hydrogen peroxide biosynthesis. The results of our proteomics analysis led to the same conclusion, as the hydrogen peroxide treatment and temperature stimuli elicited two distinct sets of responsive proteins, and the overlap was rather limited. Furthermore, none of the abundant hydrogen peroxide metabolism enzymes were changed in their abundance by increasing temperature to 29 °C. In contrast, at least several proteins in our dataset could indicate an altered temperature perception in response to the hydrogen peroxide treatment. These include a decrease in five heat shock proteins, bifunctional enolase (SMEL_009g320220.1.01), and nucleoside diphosphate kinase (SMEL_001g132570.1.01). All these candidates may reflect temperature-independent molecular processes, but an orthologue of nucleoside diphosphate kinase is a cold-responsive gene [23] and a mutant in an *Arabidopsis* orthologue of bifunctional enolase is sensitive to freezing [24].

### 3.3. Majority of Identified Temperature-Responsive Proteins is not a Part of Stress Response Mechanisms

The comparison between eggplant seed proteomes at 25 and 29 °C did not reveal any significant stress-related alterations, a fact that is well in line with the expected response of a thermophilic plant. The set of identified differentially abundant enzymes, namely pyruvate decarboxylase, isocitrate lyase, phosphoenolpyruvate carboxykinase and glutamine synthetase, indicated an increased flux in metabolism, which was likely a result of the promoted growth rate and seedling establishment. In contrast, the accumulation of desiccation-like protein SMEL_005g227570.1.01 could correspond to water deficiency. Its *Arabidopsis* orthologue was found to be accumulated in water-limited conditions [25], and one could speculate that a promoted evaporation and water consumption at 29 °C could have impacted water availability in our assay. Hydrogen peroxide via respiratory burst oxidases participates in a drought-stress response [26], and the change in desiccation-like protein average abundance at 29 °C in hydrogen peroxide-treated seeds was below the significance threshold, indicating a possible disruption in water balance sensing.

### 3.4. Hydrogen Peroxide Effect on Eggplant Seed Proteome Indicates Depletion of Hydrogen Peroxide Pool

Besides the enzymes of hydrogen peroxide metabolism, we found several candidate proteins related to redox-mediated processes, including five enzymes of glutathione metabolism and two peptidyl-prolyl cis-trans isomerases. Furthermore, there was a functional overlap with the previously found hydrogen peroxide-responsive proteins, including glycine-rich RNA-binding protein (SMEL_001g117440.1.01; *Arabidopsis* orthologues rapidly accumulated in response to hydrogen peroxide [27]), nucleoside diphosphate kinase (SMEL_001g132570.1.01; protein interactor of catalase [28]) or peptidyl-prolyl cis-trans isomerases, glutathione S-transferases, and 70s heat shock protein (hydrogen peroxide-responsive poplar proteins; [29]). Intriguingly, all these hydrogen peroxide-responsive proteins were depleted in eggplant seeds germinated in the presence of hydrogen peroxide. This could indicate that the hydrogen peroxide treatment boosted seed′s scavenging mechanisms, resulting not only in the elimination of the supplied hydrogen peroxide but also in the depletion of its endogenous pool. This scenario would explain the observed decrease in the dominant catalases, and we could speculate that the germination that promoted the effect of hydrogen peroxide was the result of hydrogen peroxide scavenging metabolism mobilization, including accelerated lipid oxidation (reflected in the depletion of an oil body-associated protein SMEL_000g077590.1.01).

### 3.5. Depletion of Heat Shock Proteins Comparison with Phytohormone-Responsive Proteomics Indicate a Putative Link to Abscisic Acid Sensitivity

Plants contain a wide spectrum of heat shock proteins (HSPs) that are divided into five groups according to their size (HSP100, HSP90, HSP70, HSP60, and small sHSP). HSPs interact with a broad spectrum of proteins and interact with pleiotropic factors involved in the signaling pathways of multiple abiotic and biotic stress responses [30]. The proteomic analyses did not show any effect of temperature, but all three sHSPs (SMEL_008g314590.1.01, SMEL_006g268200.1.01, and SMEL_006g268190.1.01) showed a significant decrease in abundance in response to the hydrogen peroxide treatment, and two HSP70s (SMEL_009g330840.1.01 and SMEL_007g271490.1.01) were depleted in the presence of hydrogen peroxide at 29 °C. The BLAST search revealed that the closest sHSP *Arabidopsis* orthologue is AT1G07400 and, based on the TRAVA (TRAnscriptome Variation Analysis) database, its corresponding gene was predominantly expressed in developing seeds [31]. Furthermore, this protein was shown to be hydrogen peroxide-responsive [32], and its overexpression resulted in abscisic acid hypersensitivity [33]. The sHSPs depletion in response to hydrogen peroxide may only represent a form of protein nutrient mobilization, but it is tempting to speculate that this protein family promotes germination rates by decreasing abscisic acid sensitivity. The interplay between hydrogen peroxide and the content of abscisic acid reportedly regulates barley seed dormancy and germination [7], and a decrease in abscisic acid content or signaling could correlate with a decrease in ethylene biosynthetic enzyme aminocyclopropane-1-carboxylic acid oxidase (SMEL_000g046320.1.01) because ethylene counteracts abscisic acid signaling pathways and induces germination [34]. To provide further support for this conclusion, we compared our hydrogen peroxide-responsive proteins to those found in response to phytohormones [35]. Altogether, orthologues for 30 proteins of our dataset have been associated with a phytohormone response, with the most represented categories being responses to abscisic acid and cytokinin with 19 and 13 proteins, respectively. Documented changes in protein abundances mostly supported a decrease in sensitivity to abscisic acid, showing abscisic acid-induced accumulation for 11 orthologues of our hydrogen peroxide-depleted proteins [36,37,38,39]. The responses of the remaining proteins were inconsistent and similar to the hydrogen peroxide effect in our dataset for eight orthologues. Furthermore, the analysis of five genes encoding hydrogen-peroxide-depleted proteins (catalases SMEL_005g238230.1.01 and SMEL_004g223080.1.01, 60S ribosomal protein SMEL_008g301560.1.01, elongation factor 2 SMEL_008g305630.1.01, and citrate synthase 2 SMEL_010g340170.1.01) revealed the presence of putative abscisic acid-responsive cis-regulatory elements. For details, see Appendix A.

## 4. Conclusions

The aim of our study was the characterization of the hydrogen peroxide and temperature-mediated promotion of eggplant seed germination rates. Seeds were imbibed in 50 mM hydrogen peroxide or mock, and they germinated for three days at 25 or 29 °C. A germination assay pointed out a similarity between the effects of hydrogen peroxide and increased temperature, but these were not confirmed in the subsequent proteome profiling. Our results suggest that hydrogen peroxide imbibition negatively impacts the endogenous hydrogen peroxide pool by mobilizing scavenging pathways and thus promoting germination. Furthermore, our data and comparison with the previous experiments indicate that small heat shock proteins could mediate hydrogen peroxide’s effect via the modulation of sensitivity to abscisic acid. Finally, our analysis resulted in the identification of numerous novel targets for detailed mechanistic studies using, e.g., mutants and transgenic plants, but this challenge is beyond the scope of this manuscript.

## 5. Materials and Methods

### 5.1. Plant Material

Eggplant seeds (Nero cultivar) used in these experiments were obtained from Moravoseed CZ a.s., Mikulov, Czech Republic. Seeds were surface sterilized, sown on a filter paper (Whatman), placed on a Petri dish (9 cm in diameter), moistened with 3.5 mL of distilled water or 50 mM hydrogen peroxide per 50 seeds, and cultivated in a growth chamber (AR36LX, Percival, 20–29 °C, darkness). The experiment was performed in at least four biological replicates, consisting of 50 seeds per replicate. The number of germinated seeds was determined by counting seeds with the emerged radicle from the seed coat.

### 5.2. Protein Extraction and LC–MS Analysis

Total protein extracts were prepared by acetone/TCA/phenol extraction [12] from ground seed tissue. The resulting protein pellets were solubilized and digested with an immobilized trypsin (Promega) overnight and desalted by C_18_ SPE [40]. Analyses were performed using a gel-free shotgun protocol based on nano-HPLC and MS/MS [41]. Briefly, tryptic digests were dissolved in 0.1% (*v*/*v*) formic acid in 4% (*v/v*) acetonitrile, their concentration was determined by a colorimetric peptide assay (Thermo), and then aliquots corresponding to 5 μg of peptide were analyzed by nanoflow C18 reverse-phase liquid chromatography using a 15 cm column (0.075 mm inner diameter; NanoSeparations (Nieuwkoop, Netherlands) and a Dionex Ultimate 3000 RSLC nano-UPLC system (Thermo, Bremen, Germany) directly coupled to a CaptiveSpray nanoESI source (Bruker) and a UHR maXis impact q-TOF mass spectrometer (Bruker, Bremen, Germany). Peptides were eluted for up to a 120 min with a 4–40% acetonitrile gradient. Spectra were acquired at 2 Hz (MS) and 10–20 Hz (MS/MS) using an intensity-dependent mode with a total cycle time of 7 s. The measured spectra were extracted by Bruker’s Data Analysis 4.1, and recalibrated MGF mascot generic format files were searched against an eggplant protein database [11] by Proteome Discoverer 2.1, employing Sequest HT, MS Amanda and Mascot 2.4 with the following parameters: enzyme—trypsin, max two missed cleavage sites; mass tolerance—35 ppm (MS) and 0.1 Da (MS/MS); modifications—up to three dynamic modifications including Met oxidation, Asn/Gln deamidation, Lys methylation, N-terminal Gln/Glu to pyro-Glu, N-terminal acetylation, and N-terminal Met loss. The quantitative differences were evaluated by calculating the normalized number of peptide spectral matches (PSM) [40]. Only proteins with at least one detectable proteotypic peptide and available quantitative data in at least two biological replicates were selected. Statistical significance was validated by a t-test (*p* < 0.05).

### 5.3. Bioinformatics Analysis

Protein-–protein interaction networks and functional enrichment were analyzed by String 11.0 [15], localizations were estimated by SUBA 4.0 [13], and protein annotations were retrieved from the UniProt database (https://www.uniprot.org/). Orthologue analyses of *Arabidopsis* genes were performed with tools at Arabidopsis information portal Araport (https://www.araport.org/) and TRAVA [31], and the analysis of the expression profiles of peroxide metabolism genes under temperature stress was performed using Geneinvestigator software, NEBION, Zurich, Switzerland [42]. Information about protein function(s) was collected from the UniProt database, the UniGene database (http://www.ncbi.nlm.nih.gov/unigene), the TAIR database (http://www.arabidopsis.org), a conserved domains search (http://www.ncbi.nlm.nih.gov/Structure/index.shtml), a homology search (http://blast.ncbi.nlm.nih.gov/Blast.cgi), the Kyoto Encyclopedia of Genes and Genomes (http://www.kegg.jp/kegg/), and the literature.

### 5.4. Data Visualization and Statistics

The statistical significance of the results was supported either by a t-test, a one-way ANOVA or the Kruskal–Wallis test (Instant Clue 0.5.4, http://www.instantclue.uni-koeln.de/; [43]). PCA analysis was performed with OriginPro 2015, OriginLab, Northampton, USA (http://www.originlab.com/).

## Figures and Tables

**Figure 1 molecules-24-04270-f001:**
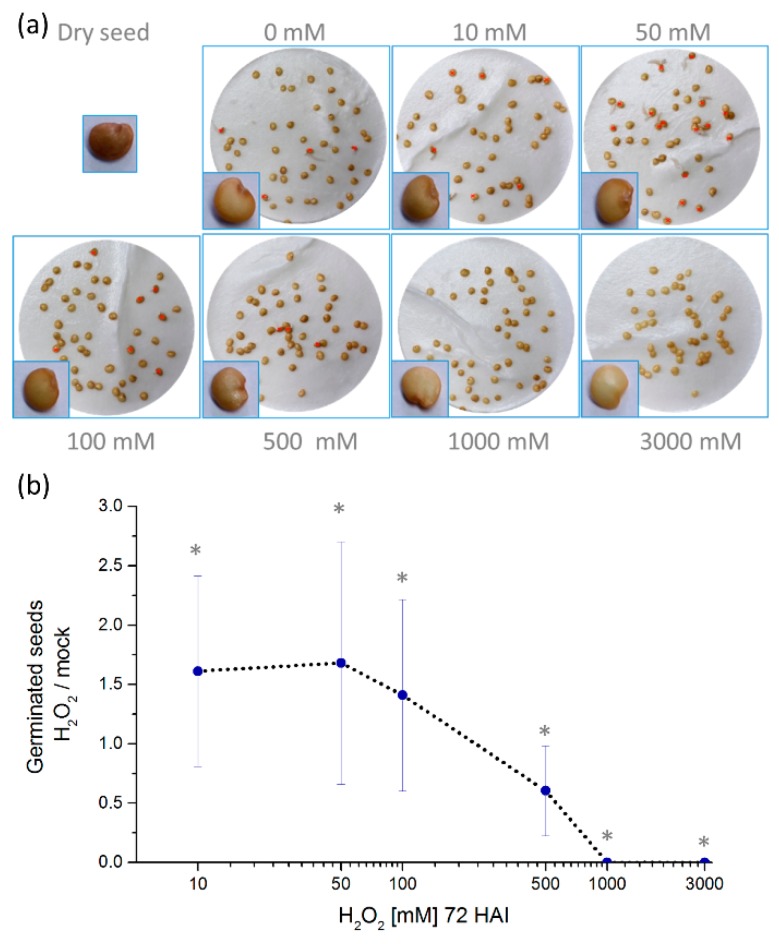
Hydrogen peroxide significantly promotes eggplant seed germination at low mM concentration. (**a**) Representative images of germination assays at 72 h after imbibition (HAI) at 25 °C. Red crosses indicate germinated seeds. **(b**) Germinated seeds at 72 HAI compared to mock. The experiment was conducted in up to eight biological replicates, each consisting of 50 seeds per plate. The average ± SD from the biological replicates is presented. * *p* < 0.05, Student’s t test.

**Figure 2 molecules-24-04270-f002:**
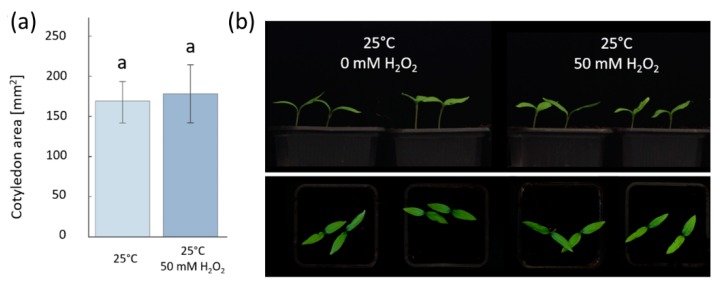
Hydrogen peroxide does not affect seedling growth. The seeds germinated after 72 h of imbibition were transferred into soil and cultivated for eight days at 25 °C. (**a**) Cotyledon area measurement and (**b**) representative photos did not show any significant difference between mock and hydrogen peroxide-treated plants (*p* < 0.05, Student’s t test). The average ± SD from the biological replicates is presented.

**Figure 3 molecules-24-04270-f003:**
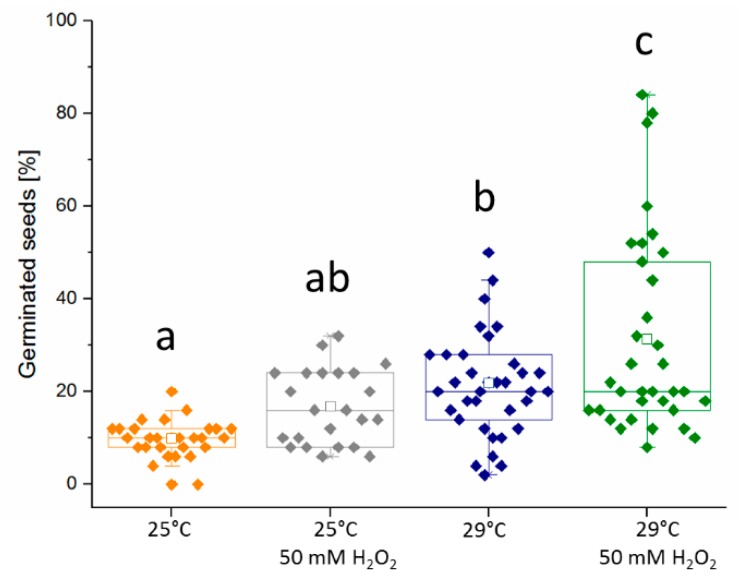
Both temperature and hydrogen peroxide promote eggplant germination. The experiment was conducted in eight biological replicates, each consisting of three independent sets of 50 seeds at 72 HAI. Letters indicate statistically significant differences (Kruskal–Wallis), and open squares represents average of all replicates.

**Figure 4 molecules-24-04270-f004:**
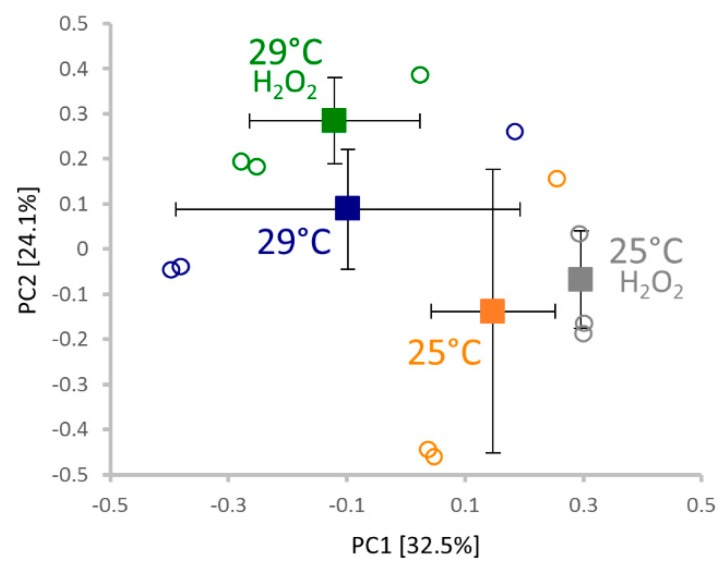
Comparison of germinating seed proteomes 72 HAI. The principal component analysis of three replicates based on normalized profiles of over 300 proteins fulfilling stringent quantitation criteria (at two unique peptides and more than 10 matched peptide spectra) revealed a clear similarity in the eggplant seed proteome composition, indicating that the effect of temperature and/or hydrogen peroxide treatment does not drastically affect major protein components. The separation of replicate averages (± SD) highlights a distinct clustering of seeds germinated at 25 and 29 °C (separated in PC1), and a less significant separation of the hydrogen peroxide treatment in PC2. Squares and circles indicate the average of replicates and individual replicates, respectively. A two-step normalization was performed to compensate for sample loading differences and equalize the contribution of lower and high-abundant proteins.

**Figure 5 molecules-24-04270-f005:**
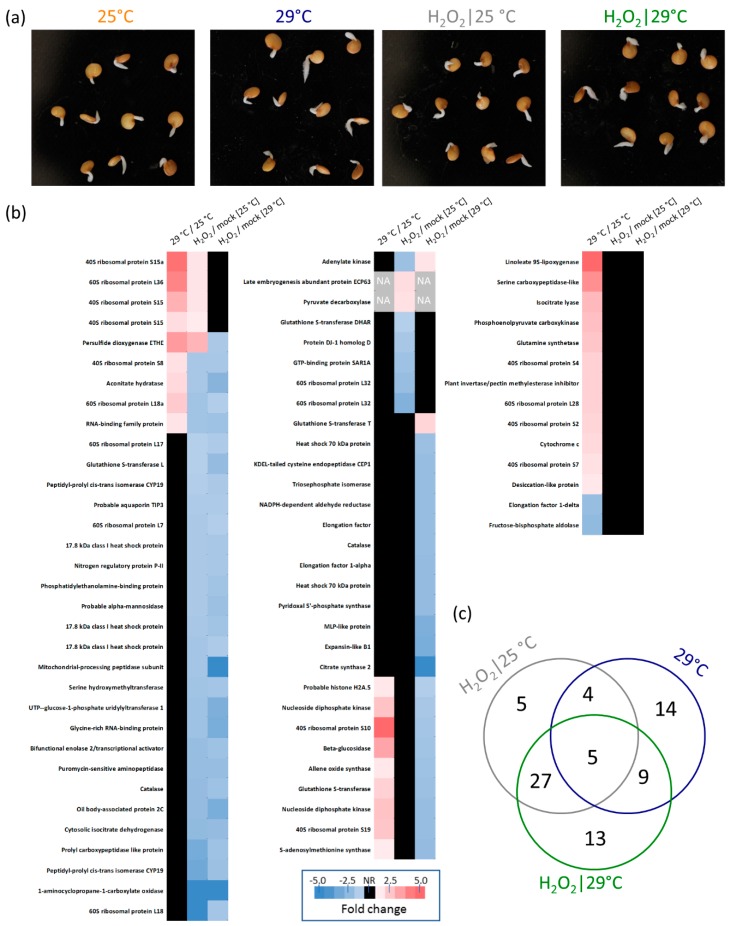
Effects of temperature and hydrogen peroxide treatment on the proteome of germinating eggplant seeds. (**a**) Representative set of germinated seeds selected for a biological replicate 72 HAI. (**b**) Differentially abundant proteins identified in response to hydrogen peroxide or temperature, with the UniProt-recommended protein name based on the closest orthologue in model plant *Arabidopsis*. Fold change represents the average of at least three replicates. See Appendix A for full details including the original protein annotations. (**c**) Venn diagram representation of the overlap between temperature and hydrogen peroxide-responsive proteins.

**Figure 6 molecules-24-04270-f006:**
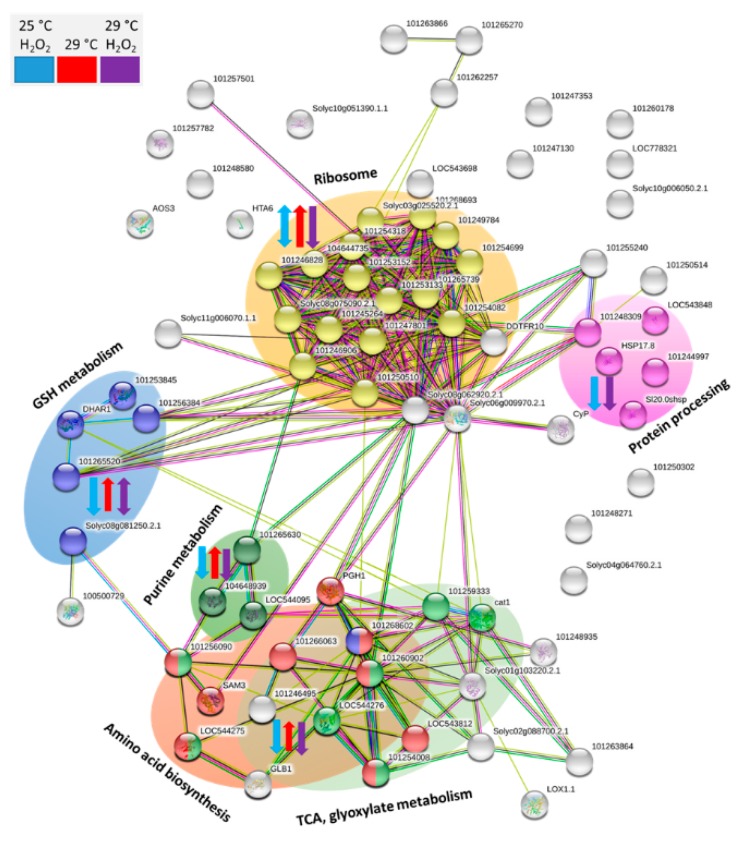
Visualization of putative interactions and processes mediated by the identified differentially abundant proteins. The network was constructed by String 11.0 [15], which employed an orthologous analysis of *Solanum tuberosum* proteins. Highlighted clusters revealed functional enrichments in the network based on biological process (gene ontology, GO) and metabolomic pathways (KEGG: Kyoto Encyclopedia of Genes and Genomes). In total, five distinct clusters are visible, including protein biosynthesis processes (ribosome and protein processing), glutathione metabolism, purine metabolism, and overlapping processes in primary and energetic metabolism (amino acid biosynthesis, TCA, and glyoxylate metabolism). Blue, red and purple arrows indicate observed changes in protein abundances for seeds incubated with hydrogen peroxide at 25 °C, seeds incubated at 29 °C, and seeds incubated with hydrogen peroxide at 29 °C, respectively.

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
