# Peer review of "Eggplant Germination is Promoted by Hydrogen Peroxide and Temperature in an Independent but Overlapping Manner"

_molecules, 2019, doi:10.3390/molecules24234270_

Round 1
Reviewer 1 Report
This study of Dufkova et al., entitled "Eggplant germination is promoted by hydrogen peroxide and temperature in an independent but overlapping manner" aims to analyze the influence of temperature and hydrogen peroxide on eggplant seeds germination rate and to correlate these data to results of differential LC-MS proteomic analysis.
Authors show that H2O2 enhances the eggplant seed germination mainly at higher (29°C) temperature, as also does the temperature itself (in comparison to 25°C). The seed imbibition with H2O2 does not have an effect on seedlings performance during the early development. Principal component analysis of the proteomic data separates the H2O2 effect at 29°C from the temperature effect, showing different proteome changes under these conditions. Authors discuss the proteome changes in the light of observed germination rates. The differential proteomes are not validated by any independent method.
The proposed study presents original data about the positive effect of H2O2 on eggplant germination, and provides valuable new knowledge with biotechnologically important information. It might be of interest of broader research community. Nevertheless, I found several substantial shortcomings, which must be improved:
I was not able to find an information about the timepoint of sample collection for the proteomic analysis. This crucial information would help readers to better understand the data interpretation.
Authors compare the differential proteomes in terms of individual accessions (Figure 5c). Considering the main conclusions, these comparisons could be complemented by more quantitative parameter. In this regard, the GO annotation analysis, or KEGG analysis could provide more reliable information.
Please clarify the difference between Fig. 1b (showing statistical significance between control and 50mM H2O2 treatment at 25°?) and Fig3 (showing that H2O2 does not significantly improve the germination rate at 25°C).
Discussion and conclusions:
I suggest to authors clearly explain and discuss the proteomic response of germinating seeds to temperature stimulus and H2O2 separately.
In my opinion the conclusions and the discussion suffers from the absence of validation experiments, which are a common practice in the current proteomic research.
Section 3.3: The assumption raised from the accumulation of desiccation like protein noted is very hypothetical. One may expect higher water consumption at 29°C, however, the disruption of water balance sensing is just hardly acceptable without any evidence, since the germination rate benefited from the temperature increase. Moreover, it is very vague to estimate the physiological status from the change of abundance of single protein.
Similarly, the decreased abundance of ox stress related proteins cannot be interpreted as a sign of depleted internal H2O2 level without any proof (see section 3.4). Authors claim the "boost of seeds scavenging mechanisms" (line 215) in response to external H2O2 treatment, while the arguments show rather the opposite (e.g. decreased catalase abundance).
Line 240 and 241: Please present the noted comparisons in the supplementary material.
The hypothesis about the interplay of ABA signalization with the observed differential proteomes may be strengthened by the search for abscisic acid responsive elements in the promotors of the genes encoding the differentially regulated proteins. Moreover, there is a vital relationship of ABA signalling with translation (doi: 10.1104/pp.110.160663).
Minor comments:
Please explain which comparisons were statistically evaluated in Fig. 1.
Line 179: Please show the Genvestigator data in the supplemental material.
Figure 3: Open squares indicating the averages are not visible in two treatments.
Line 171: Please highlight the outputs of the current study in the title of this section
Line 3.3: This title does not correspond to the discussed text
Line 3.5: Rewrite this title highlighting your findings
line 223 and 225 - add citations
Reviewer 2 Report
The manuscript "Eggplant germination is promoted by hydrogen peroxide and temperature in an independent but overlapping manner" reports the results of comparison of germination-promoting effects of low concentrations of H2O2 and temperature on the model of Eggplant. Authors revealed significant difference in protein production differences induced by H2O2 and temperature and performed the initial analysis of the observed proteins' level alterations. The data collected in this work are important for further understanding of H2O2 roles in plants and potential applications of this fundamental, available and "green" molecule in crop-production. In the reviewer's opinion, the manuscript is of good quality and deserves publication. There is only one general remark: it is not clear from the manuscript, why H2O2 effect is compared with temperature effect in the first place? Eggplant germination promotion by moving from sub-optimal to optimal temperature conditions is expectable. Hydrogen peroxide also known as potential promoter of seed germination, for example, see:
(1) Ogawa, K.; Iwabuchi, M. A Mechanism for Promoting the Germination of Zinnia Elegans Seeds by Hydrogen Peroxide. Plant and Cell Physiology 2001, 42 (3), 286–291. https://doi.org/10.1093/pcp/pce032.
(2) Wojtyla, Ł.; Lechowska, K.; Kubala, S.; Garnczarska, M. Different Modes of Hydrogen Peroxide Action During Seed Germination. Front. Plant Sci. 2016, 7. https://doi.org/10.3389/fpls.2016.00066.
(3) Szopińska, D. Effects of Hydrogen Peroxide Treatment on the Germination, Vigour and Health of Zinnia Elegans Seeds. Folia Horticulturae 2014, 26 (1), 19–29. https://doi.org/10.2478/fhort-2014-0002.
(4) Barba-Espín, G.; Díaz-Vivancos, P.; Clemente-Moreno, M. J.; Faize, M.; Albacete, A.; Pérez-Alfocea, F.; Hernández, J. A. HYDROGEN PEROXIDE AS AN INDUCER OF SEED GERMINATION: ITS EFFECTS ON ANTIOXIDATIVE METABOLISM AND PLANT HORMONE CONTENTS IN PEA SEEDLINGS. Acta Hortic. 2011, No. 898, 229–236. https://doi.org/10.17660/ActaHortic.2011.898.28.
Authors should explain why H2O2 and temperature effects are initially supposed to be related to some extent. Other minor remarks are listed below:
1) It seems like more data points may be useful in Figure 1 for H2O2 concentrations lower than 100 mM and ranges (p<0.05) are too large compared to observed differences between points.
2) Figure 4 may need more comments to make it more clear to the reader.
Author Response
We thank the referee for positive comments about our work.
There is only one general remark: it is not clear from the manuscript, why H2O2 effect is compared with temperature effect in the first place? Eggplant germination promotion by moving from sub-optimal to optimal temperature conditions is expectable. Hydrogen peroxide also known as potential promoter of seed germination. Authors should explain why H2O2 and temperature effects are initially supposed to be related to some extent.
We thank the referee for this comment. Indeed, the justification for our hypothesis that hydrogen peroxide treatment could overcome low germination rates at a suboptimal temperature is missing. Our previous bioinformatics analyses revealed a positive correlation between expression patterns of hydrogen peroxide metabolism genes and temperature-responsive genes in abiotic stress responses, and we did believe that this study could provide further evidence for interplay between temperature and hydrogen peroxide signalling. We have extended the introduction part of the revised manuscript to include this putative link between temperature and hydrogen peroxide.
It seems like more data points may be useful in Figure 1 for H2O2 concentrations lower than 100 mM and ranges (p<0.05) are too large compared to observed differences between points.
We agree that a higher density of hydrogen peroxide concentration would seem preferable and in fact, we have tested a larger set, but these results were obtained with a new method presently pending patent application and can not be disclosed in details. However, all our results indicate that the selected concentration (50mM) is optimal.
Figure 4 may need more comments to make it more clear to the reader.
Figure 4 description has been extended to improve its clarity.
Reviewer 3 Report
The present paper is devoted to a very interesting topic, namely, the effect of hydrogen peroxide on seed germination. This is very important for agriculture. Hydrogen peroxide, although a small and simple molecule, is a complete mystery. Endogenous hydrogen peroxide plays an important role in biological processes. Hydrogen peroxide can have both positive and negative effects depending on its concentration. In the paper, the authors found that hydrogen peroxide in a concentration of 50mM is able to have a stimulating effect on eggplant seeds, which belongs to the class of dicotyledons. The authors also did significant work to study the effect of hydrogen peroxide and temperature on the proteome of germinating eggplant seed, which in the future will help to effectively influence seed germination rate. The paper as a whole makes a favorable impression.
However, there is a wish for the paper. The authors should more clearly describe the results presented in Figures 4 and 6 for a better understanding of the results by the reader.
The question also arises: how will hydrogen peroxide affect the germination of seeds of plants belonging to the class of monocotyledons?
This work should be accepted for publication in Molecules.
Author Response
We thank the referee for positive comments about our work.
The authors should more clearly describe the results presented in Figures 4 and 6 for a better understanding of the results by the reader.
We have extended Figure 4 and Figure 6 description to improve the clarity of our data visualization.
The question also arises: how will hydrogen peroxide affect the germination of seeds of plants belonging to the class of monocotyledons?
Hydrogen peroxide effect on monocot germination has been described e.g. in barley (listed references by Ishibashi et al., and more details e.g. in our recent review Černý&Habánová et al., 2018), and our experiments in a cross-species comparison indicate that the key difference seems to be in the seed surface and optimal hydrogen peroxide concentration.
Round 2
Reviewer 1 Report
I would like to thank to the authors for their answers.
Unfortunately I still cannot agree with the quality of the discussion in its current state, when the main achievements of the study are substantiated by speculations. In the absence of validation experiments, the finding of new phenomena (water shortage and enhanced germination; boost of H2O2 scavenging while no change in H2O2 decomposing enzymes abundances) should be substantiated by adequate similar findings in the literature. I kindly ask the authors to explain both discrepancies in the manuscript. All data in the manuscript point rather to attenuated H2O2 scavenging capacity (at least enzymatic) - the major H2O2 scavenging enzymes are not changed (APXs, GsTs;line 140) or decreased (catalase). Which mechanism takes the responsibility for H2O2 decomposition? Please, clarify also the putative "depletion" of catalase. All catalases show rather no change or increase in abundance based on Fig. 5b. Check thoroughly all data in fig. 5B and correct the results and discussion accordingly. Please refer to supplemetary tables in the relevant places in the text.
All my other comments were addressed satisfactorily.
Reviewer 2 Report
The Authors have corrected the manuscript and now it is acceptable for publication in Molecules.